# An ultrastrongly coupled single terahertz meta-atom

Shima Rajabali [1✉], Sergej Markmann[1], Elsa Jöchl[1], Mattias Beck [1], Christian A. Lehner[2], Werner Wegscheider[2], Jérôme Faist[1] & Giacomo Scalari [1✉]

Free-space coupling to subwavelength individual optical elements is a central theme in quantum optics, as it allows the control over individual quantum systems. Here we show that, by combining an asymmetric immersion lens setup and a complementary resonating meta-surface we are able to perform terahertz time-domain spectroscopy of an individual, strongly subwavelength meta-atom. We unravel the linewidth dependence as a function of the meta-atom number indicating quenching of the superradiant coupling. On these grounds, we investigate ultrastrongly coupled Landau polaritons at the single resonator level, measuring a normalized coupling ratio $\frac{\Omega}{\omega} = 0.6$. Similar measurements on a lower density two dimensional electron gas yield a coupling ratio $\frac{\Omega}{\omega} = 0.33$ with a cooperativity $C = 94$. Our findings pave the way towards the control of ultrastrong light-matter interaction at the single electron/resonator level. The proposed technique is way more general and can be useful to characterize the complex conductivity of micron-sized samples in the terahertz domain.

[1] Institute of Quantum Electronics, ETH Zürich, 8093 Zürich, Switzerland. [2] Laboratory for Solid State Physics, ETH Zürich, 8093 Zürich, Switzerland. ✉email: shimar@phys.ethz.ch; scalari@phys.ethz.ch

Extreme electronic and photonic confinement has recently allowed a number of groundbreaking advances in several fields, from fundamental studies[1] to applications[2,3]. Particularly interesting is the possibility to manipulate on-chip the light-matter-coupling achieving new quasi-particles called cavity polaritons[4] that profoundly modify the electro-optical properties of the constituent elements.

Cavity polaritons rely on enhanced vacuum field fluctuations $\mathcal{E}_0$ to reach large values of the vacuum Rabi frequency $\Omega \propto \mathcal{E}_0$ quantifying large values of coupling[5]. The dependence on the inverse of the cavity volume $\mathcal{E}_0 \propto 1/\sqrt{V}$ led to the development of strongly subwavelength structures to reach the strongest couplings to-date[6,7]. Metallic-based cavities have been used in the mid-infrared (Mid-IR)[8] and terahertz (THz)[9] regions of the electromagnetic spectrum for strong coupling experiments. Due to their subwavelength nature, the experiments have been carried out on multiple cavities (100–1000 or more) in order to enhance the signal-to-noise ratio (SNR). Since the frequency components excited for subwavelength features cannot propagate in the far-field, only recently near-field techniques proved to be successful in probing single resonator-based strongly coupled systems at Mid-IR frequencies at room temperature[10,11]. In the field of ultrastrong coupling, the development of the Landau polariton platform[12] has led to very interesting developments, including studies of non-linear phenomena[13], the reaching of high cooperativity[14] and the study of magnetotransport[15] in ultra-strongly coupled Hall systems[16]. In this approach, extremely subwavelength planar split-ring resonators have been used in combination with one-dimensionally confined semiconductor heterostructures to achieve normalized coupling values $\frac{\Omega}{\omega} > 1.4$ (ref. [17]).

The light-matter interaction strength in most of the solid-state-based cavity quantum electrodynamics (QED) systems scales with $\sqrt{N}$, $N$ being the electron number due to the collective (Dicke) enhancement of the interaction strength[18]. In Landau polaritons, when employing metasurface cavities, the number of coupled electrons is shared among several hundreds of identical cavities. In the last few years, there has been a growing interest in studying ultrastrong light-matter coupled systems toward the limit of few electrons coupled to a single cavity[19–21] to study the fermionic Rabi model for the coupled systems rather than the bosonic Hopfield description where both material excitation and the electromagnetic field are described as boson fields. Moreover, such systems could be building blocks for performing quantum information tasks. In this perspective a reduction in the number of coupled carriers toward the low limit of a few and ultimately a single coupled element constitutes a great challenge both in terms of measurement sensitivity[1,22] and optical cavity design.

The possibility and the limits on the interaction of free propagating far-field optical beams from individual elements (molecules, atoms, quantum dots, etc.) have been investigated both theoretically[23,24] and experimentally[25,26], mainly at visible or infrared frequencies.

In this work, the single resonator spectroscopy at sub-THz/THz spectrum is performed by implementing a simple and practical asymmetric Silicon immersion lens (aSIL) configuration. Here, the individual element is not a quantum object but rather a single-subwavelength metallic planar resonator, operating at millimeter wavelengths, lying in-between free space propagating beams and guided microwaves. We also demonstrate the linear dependence of the cold cavity linewidth of a two dimensional (2D) array of the metasurface on the number of resonators in the array when the illuminated area with resonators becomes comparable or smaller than the wavelength. This observation indicates the quenching of the superradiance decay. Moreover, the

Landau polaritons in a coupled single-subwavelength resonator are resolved on two different semiconductor heterostructures, and high normalized coupling strengths of up to 60% and high cooperativities of up to $C = 94$ are achieved.

## Results

**Single resonator detection and linewidth.** Free-space probing of a sample containing a few/single-subwavelength resonators is difficult as it is intrinsically inefficient in exciting the right electromagnetic modes and generally features a very low SNR. We thus developed a strategy based on the employment of a complementary metamaterial combined with an aSIL configuration to optimize the matching of a free space propagating beam to a single element of a complementary metasurface. The conventional THz-time domain spectroscopy (TDS) setup we employ has a pair of off-axis parabolic mirrors first to collimate and then focus the incident THz beam from the photoconductive switch[27] on the sample. Afterwards, the transmitted signal from the sample will be collected, collimated, and then focused on the zinc telluride (ZnTe) crystal through another pair of off-axis parabolic mirrors (Fig. 1a). The detection is performed using an electro-optic detection scheme[28]. The beam in this setup, that we can assume to have a Gaussian profile, has a spot size of around 3 mm at the sample's location. In order to correctly couple the incoming THz beam to the single complementary meta-atom (Fig. 1b, c), we designed a back-to-back lens system inspired by some previous studies on single quantum dots at visible wavelengths[26] (Fig. 1d). Being very akin to a $4f$ arrangement, a similar setup has been recently proposed also as a THz spatial filter[29]. A set of ray optics simulations in COMSOL Multiphysics is done to find the correct lens dimensions and adjust the input and output numerical aperture of the system such that it is compatible with the existing THz-TDS setup. A final design consists of a front hyperhemispherical Silicon (Si) lens of diameter 4 mm that focuses the incident beam to a spot size of ~500 μm at the surface of the sample (Fig. 1e), where the metamaterial resonators are located. The optical images of the setup are provided in the Supplementary Material (Fig. S1). The back hemispherical Si lens of diameter 8*mm* collects the transmitted beam from the sample and sends it through the off-axis parabolic mirrors to the detector. To optimize the dimensions for the back lens, we run a parametric sweep on the back lens diameter in the COMSOL simulation. The goal was to match the output numerical aperture of the lens system with the numerical aperture $f\#3.85$ of the off-axis parabolic mirror according to the chosen thickness and material for the substrate at the operating frequency ($f = 300$ GHz). In our case, we used a 500 μm thick semi-insulating gallium arsenide (GaAs) slab that corresponds to the average thickness of the substrates used for the epitaxial growth of the samples. The presence of the substrate where the sample is deposited also makes the back lens effectively a hyperhemispherical one. The significant advantage of this asymmetric design compared to the previous symmetric designs[26,29] is that there is no need to sandwich the target surface (in our case, the metamaterial metallic thin film) between two semiconducting substrates to have the surface at the focus point of the confocal system. The asymmetry of the lenses accounts for the sample substrate thickness as an active part of the optics. The sample itself is located on top of the substrate and has a very small thickness compared to the wavelength. The planar complementary metasurface and the underlying quantum well (QW) sum up to less than 1 μm in the propagation direction of the THz beam. Thus the back lens with a larger diameter compensates for the substrate thickness and collects the diverging beam at the back interface of the substrate. The performance of our aSIL setup

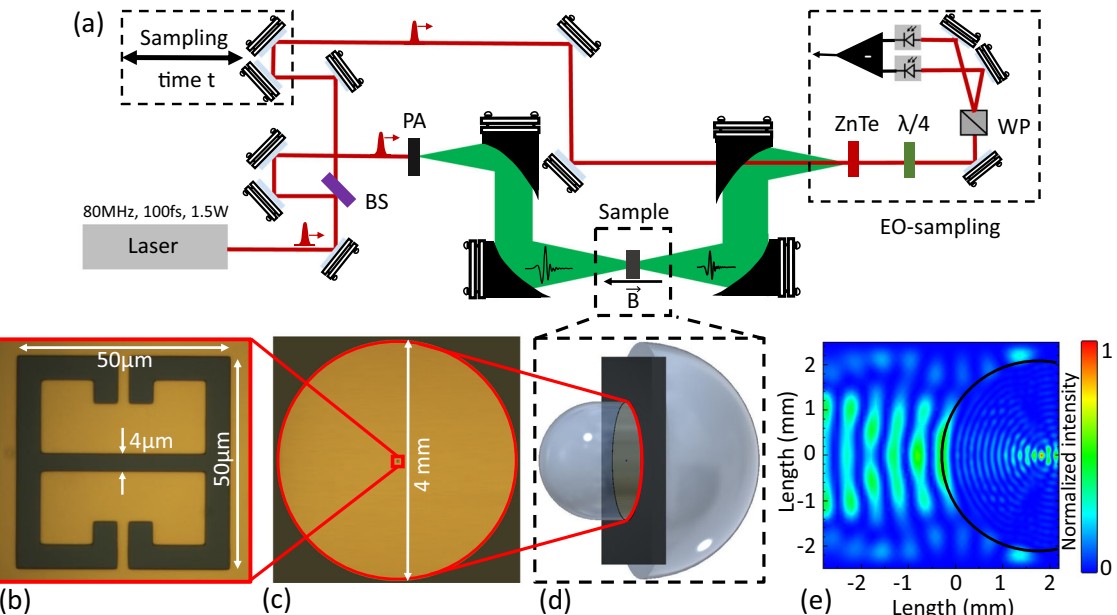

**Fig. 1 Schematic of the THz-TDS setup with the aSIL configuration. a** The schematic sketch of the THz-TDS used for the transmission experiments. BS, PA, λ/4, and WP stand for beam splitter, photo-conductive antenna, quarter waveplate, and Wollaston prism, respectively. **b** The optical image of the single resonator including the periodicity and gap size. **c** The optical image of the whole sample metallized with a circular boundary of diameter 4 mm for the alignment of the front lens. **d** The magnified schematic of the aSIL configuration. **e** The COMSOL simulation result of the focused THz beam at the interface between the front lens and the resonator field.

could in any case be further improved by a careful choice of the lens curvature to optimize the matching with the incoming THz beam and by providing the Si lenses with a broadband anti-reflection coating to further enhance the SNR[29].

To-date, single meta-atom spectroscopy has been possible only employing near-field techniques[10,11,22], that are difficult to implement in our experimental condition, i.e., low temperatures and high magnetic fields. Here we employ the back-to-back aSIL setup to study a single complementary resonator. We previously demonstrated the efficiency of the complementary THz meta-surfaces in the context of Landau polaritons[30]. The significant advantage of this configuration in the transmission measurement of a single resonator is that the measured signal above the noise level is directly the signal of interest yielding a very high contrast. In the case of a standard or direct metasurface (highly transmissive sample), the challenge of the measurement is to detect a very small absorption change on top of a large signal. If one employs a complementary metasurface, the situation is reversed as the transmitted signal is very low, but the only signal detected is the relevant one. Figure 2 visualizes the reasons that single resonator spectroscopy is possible using an aSIL config-uration. The front hyperhemispherical Si lens with a refractive index of $n_r = 3.42$ at 300 GHz[31] permits to optimally focus thanks to high numerical aperture and high refractive index. Both front and back hyperhemispherical lenses expand the accessible wave vectors for far-field propagation. The back lens improves the collection of the signal and increases the frequency resolution due to refractive index matching with the sample substrate and consequent suppression of Fabry-Pérot resonances (echoes in the time domain) from the interface between the backside of the sample and the back lens.

In Fig. 2 we clarify, using a Fourier optics argument and some experimental measurements, the mechanism at the basis of the different signals observed when probing a metasurface with a large number of resonators (Fig. 2a) and a single meta-atom with a plane wave (Fig. 2b) compared to the case of a single meta-atom illuminated with an immersion lens (Fig. 2c). The meta-atoms

constituting the planar metamaterial are strongly subwavelength by design: they can be usually probed by far-field optics because the near-field components diffracted by each element are scattered in the far-field by the surrounding elements of the array. The unit cell dimension $d_{unit}$ is highly subwavelength as well (in our case $d_{unit} = 70\,\mu m$ for a free space wavelength $\lambda_{fs} \simeq 1\,mm$, with $\frac{d_{unit}}{\lambda_{fs}} = 7 \times 10^{-2}$) allowing the spatial frequencies to propagate as represented in Fig. 2a and its corresponding experimental far-field spectrum in Fig. 2d, as usually observed in complementary metamaterials[32]. The presence of the resonator array is as well required for the excitation of the resonant mode as the nearby elements diffract the incoming wave providing some in-plane components of the wave vectors. In the case where the plane wave illuminates a single-subwavelength meta-atom (Fig. 2b), a large portion of spatial frequencies remain trapped in the resonator's near-field and do not reach the far-field detector. The corresponding experimental spectrum does not show the resonant feature at 300 GHz (Fig. 2e). When the aSIL arrangement is used (Fig. 2c), the spread in the wave vector excites the correct resonator mode that can be then collected by the back lens to recover the searched transmission resonance as experimentally shown in (Fig. 2f). In the lower row of Fig. 2, we plot the near-field distribution of the electric field (Fig. 2g) and its spatial Fourier transform (Fig. 2h). The circles in Fig. 2h, correspond to the maximum amplitude of the wave vectors propagating in the cases without lenses (blue) and with lenses (red). It is evident that with the lenses the quasi-totality of wave vectors can propagate in the far-field. As a further confirmation of our model, we can cite the results obtained with a near-field probe measuring a similar complementary single resonator[22]. In that experiment, the single resonator was excited with a focusing lens, providing the necessary in-plane wave vectors. The resonant signal was found to remain trapped in the near-field since there was no collecting lens. Now that we clarified the mechanism that underlies the observed transmission spectra, we can discuss the experimental measurements in detail.

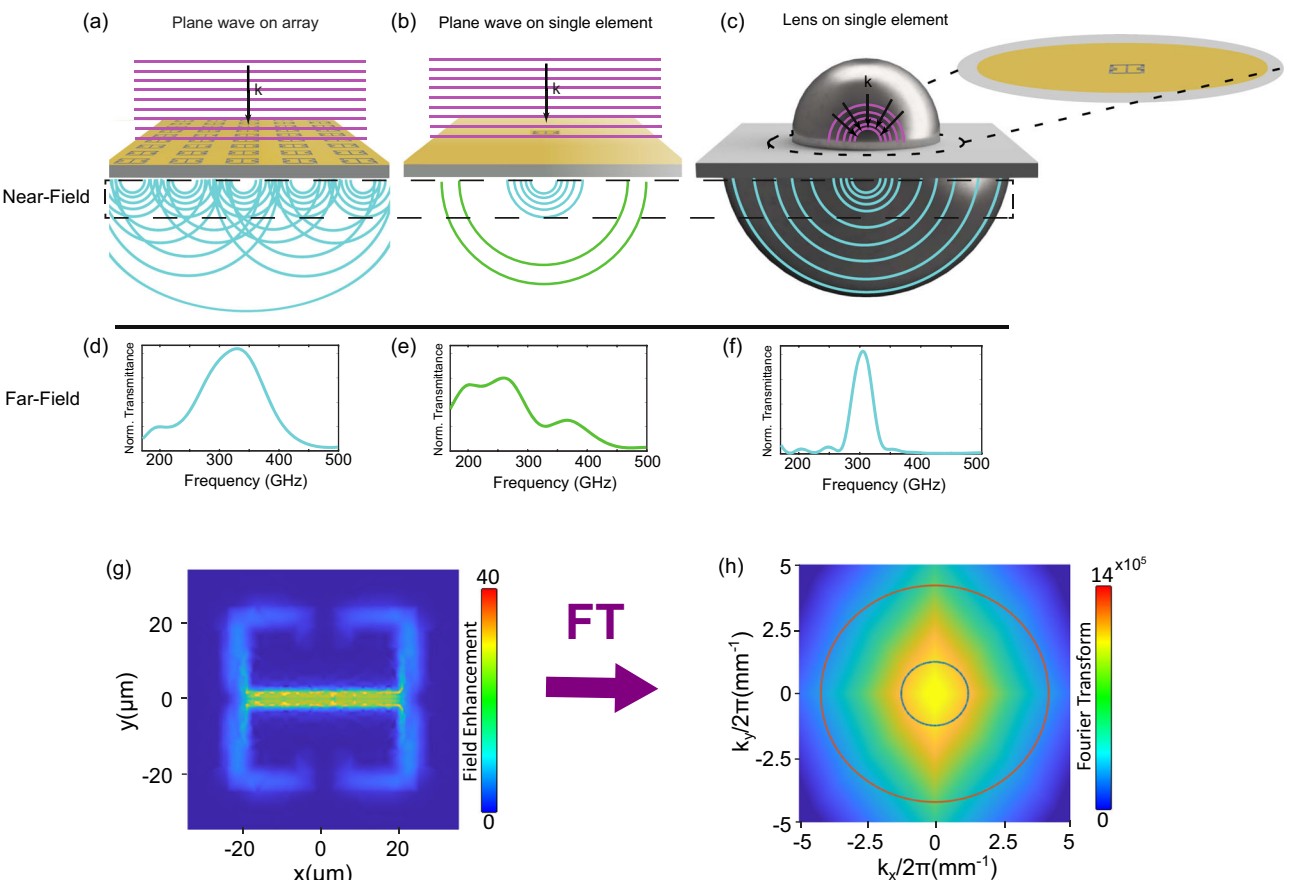

**Fig. 2 No lens vs. aSIL configuration for single resonator spectroscopy.** A schematic of the near-field THz-resonator interaction in the case of an array without lenses (**a**), a single resonator without (**b**) and with (**c**) lenses. The magnified view in **c** displays the single meta-atom within the circular boundary for the alignment at the interface with the front lens. Experimental far-field measurements (**d**–**f**) corresponding to the configurations sketched in **a**–**c** of an array without lenses and a single resonator in case of without and with aSIL configuration. **g** The finite element simulation result of the field enhancement at the resonator plane and at the resonant frequency for a single resonator excited with an incident plane wave with a polarization along y-axis. **h** 2D Fourier transform of the field in **g**. The blue and red circles represent the accessible wave vectors for far-field propagation in case of having incident waves in the air (without lenses) and in Si (with the aSIL arrangement), respectively.

As a first step, we perform a systematic study of the cold cavity, illuminating a series of samples deposited onto a semi-insulating GaAs substrate. These samples contain a decreasing amount of resonators, from 3600 down to a single one. We study the linewidth and the transmission amplitude of the cold cavity and their dependency on the number of resonators. The theory of the superradiance[18,33–35] predicts a reduction in the collective radiative decay and consequently an improvement in quality factor (Q-factor) by reducing the number of two-level emitters confined in a spatial region comparable or smaller than the wavelength. This has also been investigated experimentally on metamaterials but never in the single resonator limit[36]. In order to study this effect, we designed and fabricated 2D array of $n \times n$ complementary split-ring resonators (cSRR), including a very large array of $60 \times 60$ resonators and small arrays with a varying $n$ from 6 to 1, operating at a resonant frequency of 350 GHz and a periodicity of 70 μm on a GaAs substrate (Fig. 1b).

We measured the resonators with a commercial THz spectrometer (details in "Methods" section) at room temperature. We conducted the measurements with and without the back-to-back aSIL system. As expected from previous research[22], the resonant peak of the $2 \times 2$ cSRR array and the single resonator could not be resolved without the use of our aSIL arrangement because of the weak interaction of a few resonators with the THz beam (see the Supplementary Material, Fig. S2). Even the measurement with nine resonators shows a very

weak resonant peak. In Fig. 3a, we report the result of the study, showing the collected spectra using the aSIL assembly and the extracted Q-factors for the measurements with lenses. The measurement shows about five times signal enhancement in the peak-to-peak value of the THz waveform in the time domain for the single resonator (see Supplementary Material, Fig. S3). Moreover, the resonant peak for the arrays with less than nine resonators were resolved with a large dynamic range (60 dB for the single resonator).

Another key feature of this technique is the significant improvement in frequency resolution. Since Si and GaAs have very close refractive indices up to 1 THz[31], the echoes of the THz pulse, reflected at each interface, are greatly suppressed due to the impedance matching and, as a consequence, the etalon effects are minimized. Particular care was taken to assemble the stack lens-sample-lens without any air gap, pressing on the assembly with metallic positioners. In the best conditions, the resolution was 18 GHz (corresponding to a time scan length of 55 ps), limited by the reflection from the detector. If we compare the results of the measurements on the large arrays with and without the lenses we observe a difference in the resonant frequency due to the presence of the front Si lens. The presence of an high index material in the near-field of the complementary resonators red-shifts the resonance to values below 300 GHz which is also confirmed by our finite element simulations (more information in the Supplementary Material, Fig. S4).

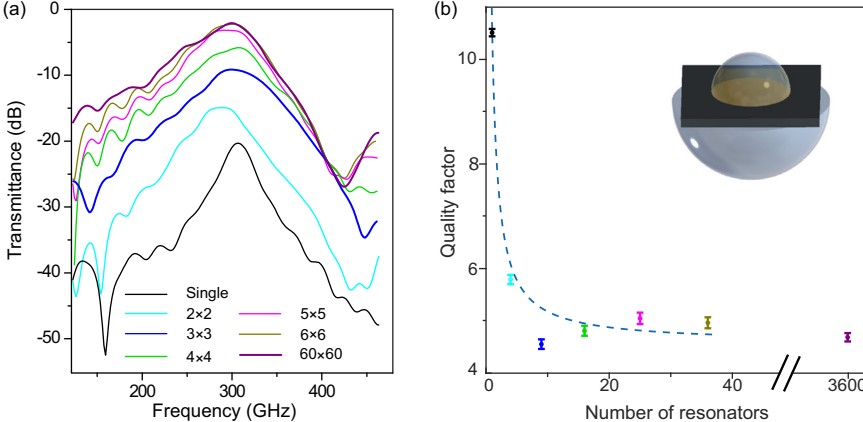

**Fig. 3 Quenching of the superradiance decay. a** Transmittance of $n \times n$ array of cSRRs ($n = 1\text{–}6$) and a large array with $n = 60$. These measurements are done using the aSIL assembly and the resonant peak of the single resonator is resolved. **b** is Q-factor vs. the number of resonators extracted from the measurements with lenses. The color of the data points are the same as the color of the transmittance curves in **a**. The blue dashed line shows the linear dependence of the quality factor on the density of the resonators ($Q = \frac{6.04}{n^2} + 4.6$). The Q-factors are calculated by fitting the resonant peak with Breit-Wigner-Fano function. All error bars represent one $\sigma$ confidence interval.

To calculate the Q-factor, we fitted the resonant peak with a Breit-Wigner-Fano function. Figure 3b shows an almost constant Q-factor for 3600 to 16 resonators measured with lenses. For the samples with fewer resonators ($n \leq 3$), the dimension of the resonator field becomes comparable to the wavelength (at 300 GHz, the wavelength propagating in GaAs is $\lambda_n = \lambda/n_{GaAs} = 278\,\mu m$ and the one propagating in Si is $\lambda_n = \lambda/n_{Si} = 292\,\mu m$), causing the quenching of superradiance to emerge. The Q-factor doubles from nine resonators to a single resonator. We attribute this Q-factor enhancement to the quenching of the superradiant decay rather than to inhomogeneous broadening that is quite unlikely in this kind of THz resonators. The Q-factor reaches ~11 for the single one. This can have important consequences in the context of the strong coupling experiments that we will discuss in the next section. If we compare our results with similar studies performed on arrays of metamaterials down to a few elements[37], we observe a similar behavior as in the case of "incoherent" resonators that are not coupled by magnetic interaction, since the symmetric arrangement of the inductors in our case cancels out the magnetic response. What emerges from our data is the enhancement of the Q-factor when the area occupied by the resonators is smaller than $\lambda^2$. It is worth noting that the number of illuminated resonators in case of the large array ($60 \times 60$ resonator) are 2628 for the measurement without lenses (a 4 mm diameter illuminating spot) and 44 for the measurement with aSIL configuration (a 500 $\mu m$ diameter illuminating spot).

**Ultrastrong coupling in a single complementary resonator.** After having characterized the measurement technique and studied the Q-factor dependence of the cSRR array on the number of resonators, we apply the developed method to the spectroscopy of a single resonator ultrastrongly coupled to inter-Landau level transitions in a single GaAs QW. With a similar fabrication process, a large array and a single cSRR were deposited on top of a two dimensional electron gas (2DEG) produced in a single GaAs square QW located 90 $nm$ below the surface. The sample transmission is then measured in a THz-TDS setup at cryogenic temperature ($T = 2.7$ K) as a function of magnetic field swept between 0 and 4 T. The best resolution for this setup is 33 GHz (corresponding to a time scan length of 30 ps), limited by the thickness of the cryostat windows. Further details are to be found in the Methods section. In order to gain insight into the coupling of the complete metasurface and the single resonator to the THz beam, we measured the transmission spectra for both kinds of

samples with and without the aSIL assembly. The results, displayed as transmittance colormaps as a function of the magnetic field, are reported in Fig. 4a. The figure shows a comparison between the transmission measurements without (top row) and with aSIL configuration (bottom row) for a 2D array of $60 \times 60$ resonators (left column) and a single resonator (right column). The colormaps relative to the plane wave (without lenses) case clearly show very well-resolved polaritonic branches for the 3600 resonator sample and a very different spectrum in the case of a single resonator. In the single resonator measurement, we observe a broad spectral feature corresponding to high transmission that extends from 180 to 300 GHz and is independent from the applied magnetic field. A narrow absorption feature linearly changing with the magnetic field and corresponding to the cyclotron resonance crosses the broad transmission peak. The single resonator excited by a plane wave, whose radiation is collected without an immersion lens, does not show ultrastrong coupling since the correct resonant LC mode cannot be excited and detected. The measurement in the case of the single resonator measured with the aSIL assembly is significantly different: we observe extremely well-resolved polaritonic branches that compare well with those observed in the 3600 resonator sample in both cases with and without the lenses. We can now analyze in detail and compare the polaritons measured in the case of employing the aSIL assembly for the single resonator and the large array.

The lower polariton (LP) mode of the coupled single resonator at its asymptotic limit, at $B = 4$ T, has a Q-factor of 15.4 (using time trace decay method, more information in the Supplementary Material, Fig. S7). The Q-factor of the LP at $B = 4$ T for the sample with the coupled 2D array of 3600 resonator without lenses and with lenses are 3.4 and 6.9, respectively. Similar to the cold cavity in the previous section, the Q-factor of the LP at its asymptotic limit is higher for the single resonator compared to the one for the array due to the quenching of the superradiant decay. By extracting the maximum of the spectrum at each magnetic field and fitting them with the Hopfield model[5], a normalized coupling of $\frac{\Omega}{\omega} = 33\%$ is achieved for the single resonator measured in a confocal configuration. For the array of 3600 resonators the normalized coupling for the measurement with and without lenses are $\frac{\Omega}{\omega} = 32\%$ and 36%, respectively. The cross sections of the colormap of the $60 \times 60$ array and the single resonator measurement with lenses at three different

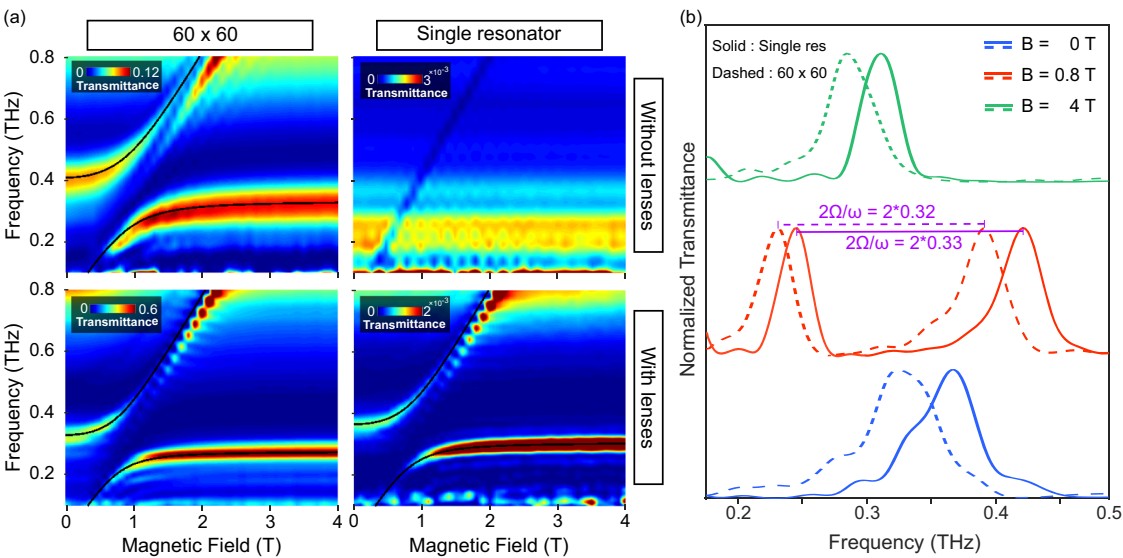

**Fig. 4 Comparison between THz-TDS measurements of a large array and a single cSRR coupled to LL transitions in a GaAs QW with/without Si lenses. a** The transmission measurements without (top row) and with the aSIL configuration (bottom row) for a 2D array of 60 × 60 cSRRs (left column) and a single resonator (right column). The polariton branches in the measurement of the single resonator without lenses are not resolved due to the weak interaction of the resonator with the THz beam. The black solid lines are the fitted LP and UP using the Hopfield model. **b** Sections of the colormap for the single resonator (solid lines) and 60 × 60 array (dashed lines) both measured with lenses at different magnetic field values ($B = 0$, 0.8 T, (anti-crossing) and 4 T) to demonstrate the well-resolved polariton branches. The distance between the peaks at $B = 0.8$ T is the vacuum Rabi splitting (marked with purple) and is twice of the coupling strength. All the modes are normalized to their maximum value. The vertical shift is added for clarity.

magnetic field values of 0 T, 800 mT, and 4 T show the well-resolved upper polariton (UP) and LP peaks in the single resonator measurement (Fig. 4b). It is also evident that there is a renormalization of the loaded cavity frequency that blueshifts the resonance by 30 GHz ($\simeq 0.1\,\omega$) when we reduce the number of resonators from 3600 to 1. Interestingly, the normalized coupling strength remains basically the same ($\frac{\Omega}{\omega}|_{single} = 0.33$, $\frac{\Omega}{\omega}|_{3600} = 0.32$). We observe a much smaller shift (7 GHz) when we investigate the cold cavity at 300 K (Fig. 3a and Fig. S5 in the Supplementary Material). We attribute the larger shift in the case of the loaded cavity to the substantial dielectric contribution of the 2DEG that effectively produces a slow light effect, enhancing the collective Lamb shift of the ensemble of meta atoms as observed in other systems[38–40].

The calculated cooperativity for the coupled single resonator measurement is equal to $C = 94$ and for the large array of resonators are $C = 26.4$ without lenses and $C = 37$ with the aSIL configuration. The cooperativity is calculated using $C = \frac{4\Omega^2}{\kappa\gamma}$ where $\Omega$, the coupling strength, is half of the mode splitting at the anti-crossing, $\kappa$ is the mode dissipation (cavity decay) rate and $\gamma$ is the decoherence rate. $\kappa$ and $\gamma$ are both extracted from the direct measurement of the cyclotron coherence time[41] using TDS measurement of the 2DEG measured with the confocal system (more information in the Supplementary Material, Figs. S6 and S7). The linewidth of the cyclotron resonance is smaller than the one measured without lenses due to the quenching of the superradiance effect that has been proven to be the intrinsic limit[42]. In our case the effect of the lens can be understood in a similar fashion to what observed in the case of superradiantly broadened intersubband plasmons[43]. An estimate (similarly to ref. [20]) of the number of optically active electrons at the top most LL for our single cavity with a cavity surface of $S = 155\,\mu m^2$ on a single GaAs QW yields $n = \frac{eB}{\hbar} \times S = 2.42 \times 10^{14}[\frac{1}{m^2T}] \times B \times S \simeq 30,000$.

In order to increase the coupling and reduce the number of optically active electrons, we employ an indium antimonide (InSb)-based 2DEG[44,45] at the same distance from the surface as

the GaAs 2DEG (90 nm). Due to the lighter effective mass of electrons in InSb ($m_{eff} = 0.0225m_0$), the anti-crossing for the coupled system lies at a much lower magnetic field value (at 250 mT compared to GaAs at 800 mT) for the same resonator frequency of 300 GHz (Fig. 5a). Therefore, the number of coupled electrons is reduced more than three times with respect to the GaAs case. To further reduce the cavity volume and increase the field confinement, a single cSRR with a narrower gap of 1 μm at the same resonant frequency (300 GHz) is also fabricated and measured (Fig. 5b). To enhance the coupling, this cavity is deposited onto a shallower InSb 2DEG with the same QW thickness at a distance of 50 nm below the surface. The cavity surface, evaluated with finite element simulations, is $S = 138.32\,\mu m^2$ for the larger gap and $S = 28.8\,\mu m^2$ for the smaller gap. The number of coupled electrons at the anti-crossing are then $n_e = 8368$ and $n_e = 2090$[20] for the resonators with the gap size of 4 and 1 μm, respectively.

If we extrapolate this result and consider the case of our previously demonstrated highly confined, hybrid dipole antenna split-ring resonator[20] the use of an InSb QW can lead to an ultrastrongly coupled system with less than five coupled electrons.

Due to a higher absorption in InSb QWs, the transmitted signal is less in the single resonator measurement on a GaAs-based QW. The LP mode at its asymptotic limit (at $B = 4T$) has a Q-factor of 6.4 in both cases. The normalized coupling strengths are increased and equal to $\frac{\Omega}{\omega} = 47\%$ and 60% for a single cSRR with 4 and 1 μm gap (more information about the fitting in the Supplementary Material, Fig. S8). The cooperativity in the case of these InSb based samples are $C_{gapsize:4\,\mu m} = 12.5$ and $C_{gapsize:1\,\mu m} = 14$. The lower cooperativity (compared to the values for a single resonator on a single GaAs QW) despite having a higher coupling strength is due to an order of magnitude lower mobility in InSb QWs which affects the decoherence rate (larger $\gamma$).

The cross sections of the colormaps of the single resonator on InSb QWs measured with lenses at three different magnetic field values of 0 T, 425 mT, and 1.5 T show the well-resolved UP and

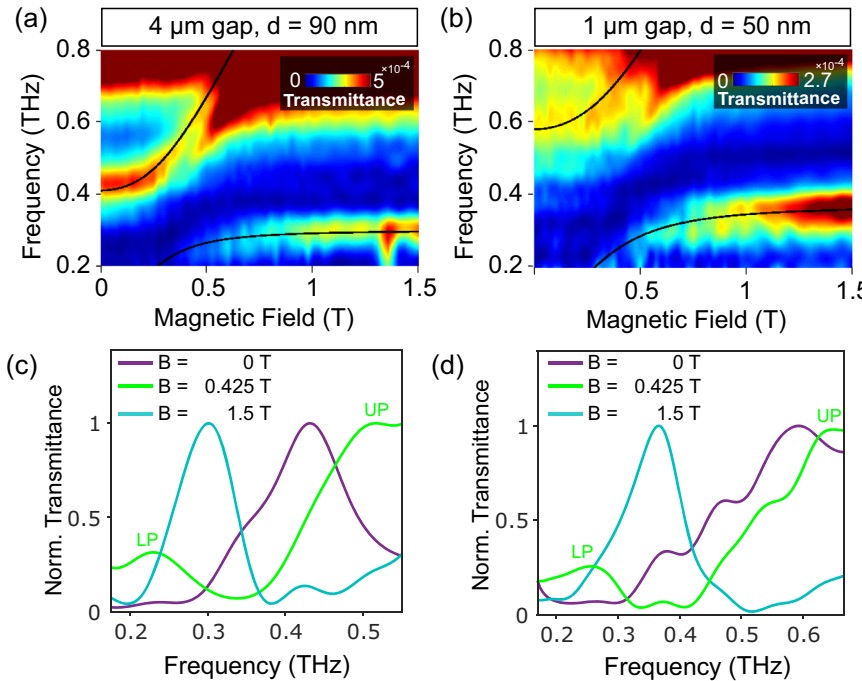

**Fig. 5 THz-TDS measurements of LL transitions in a single InSb QW coupled to a single resonator.** The colormaps correspond to the resonators with a gap size of 4 μm on an InSb QW located at a distance of 90 nm below the surface (**a**) and a gap size of 1 μm on a shallower InSb QW located at a distance of 50 nm below the surface (**b**), measured with lenses. The colormaps in **a** and **b** show normalized coupling ratios of $\frac{\Omega}{\omega} = 47\%$ and 60% and cooperativities of $C = 12.5$ and 14, respectively. The black solid lines are the fitted LP and UP using the Hopfield model. Sections of the colormaps in **a** and **b** are represented in **c** and **d**, respectively, at different magnetic field values ($B = 0$, 0.425T and 1.5T). To have a more visible LP and UP, a section at $B = 0.425$T which is at a marginally higher value than the anti-crossing is chosen.

LP peaks in the single resonator measurement (Fig. 5c, d). Since UP and LP are not both visible at the anti-crossing (~250–300 mT), a cross section at a slightly higher value at $B = 425$ mT is represented. It is worth noting that in the narrower gap resonator, the broadening of UP and partial disappearance of LP at low values of the magnetic field are ascribed to their coupling to a continuum of magnetoplasmon excitations as recently discussed in ref. [46].

To conclude, we presented a back-to-back Si immersion lens setup with an asymmetric configuration allowing the spectroscopy of highly subwavelength individual THz cavities. Using this platform, we resolved the far-field transmission measurements of an ultrastrongly coupled, subwavelength split-ring single resonator to a LL transition in a single GaAs QW and a single InSb QW. The highest coupling of 60% for only about 2000 coupled electrons is reported for a single cSRR on a single InSb QW. As our results demonstrate, the combination of the aSIL configuration with a complementary-based resonant metallic structure paves the way to single-object, highly subwavelength spectroscopy of QED systems operating in the mm-wave and THz range. The proposed experimental scheme can be extended to the study of dynamical optical conductivity of high-quality 2D structures (graphene, TMDc's, Van der Waals heterostructures)[47] with very small effective areas (i.e., $10 \times 10$ μm$^2$) resulting from exfoliation procedures.

The data that support the findings of this Article are available in the ETH Research Collection[48].

## Methods
**Asymmetric lens setup and sample fabrication**. The lenses are hyperhemispherical and hemispherical ones fabricated with high resistivity Silicon (Tydex[49]) of diameter $2r_1 = 4$mm and $2r_2 = 8$ mm, respectively. The sample is metallized with a circular boundary with a diameter of 4 mm, matching the edge of the top lens (see Fig. 1c). The lens and the sample can be then accurately aligned under the optical microscope. Mechanical clamps ensure a close contact of the whole assembly front lens-sample-back lens, forming a quasi-index matched ($n_{\text{Si}}^{350\text{GHz}} = 3.42$, $n_{\text{GaAs}}^{350\text{GHz}} = 3.52$) stack of total length $L_s \simeq 6.5$ mm. The resonators were simulated using a commercial software package (CST microwave studio) and fabricated with a direct laser writing lithography with Heidelberg DWL66+ followed by deposition of 5 nm titanium and 200 nm of gold and a lift-off process.

**THz-TDS spectrometer**. The measurements of the Q-factor of the cold cavity as a function of the number of resonators have been carried out with a commercial, fiber coupled THz-TDS spectrometer (Menlo Terasmart[50]). THz radiation is produced and detected by a pair of indium gallium arsenide-based photoconductive antennas. The beam is guided with TPX lenses with focal length of 50 mm. The whole optical path is contained in a nitrogen purged box. The measurements with the applied magnetic field are performed with a home-made THz-TDS system based on a titanium:sapphire ultrafast ($\tau < 100$ fs) laser (Mai Tai, Spectra Physics[51]) that illuminates (600 mW average power) an interdigitated photoconductive antenna[27]. The THz beam is coupled to a Spectromag cryostat (Oxford Instruments[52]) equipped with an 11 T superconducting magnet and crystalline quartz (z-cut) windows . Two mirrors (197 mm focal length, 2" diameter) focus the THz signal at the center of the superconducting coils. A variable temperature insert is used to cool down the sample at $T = 3K$. The THz signal is then detected via electro-optic sampling in a 3 mm ZnTe crystal.

## Data availability
The numerical simulation and measurement data that support the plots within this paper are available from the corresponding author upon reasonable request.

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

## Acknowledgements

G.S. would like to thank L. Novotny, R. Singh, F. Helmrich for discussions and M. Frimmer and R. Colombelli for careful reading of the manuscript and insightful comments. We acknowledge financial support from ERC Grant No. 340975-MUSiC and the Swiss National Science Foundation (SNF) through the National Centre of Competence in Research Quantum Science and Technology (NCCR QSIT).

## Author contributions

G.S. and S.M. conceived the aSIL configuration. S.R. fabricated the samples. S.R. and E.J. performed the measurements. M.B., C.A.L., and W.W. performed the epitaxial growth. Data interpretation and discussion was made by J.F., G.S., S.R., E.J. and S.M. G.S. and S.R. wrote the manuscript. All the work was performed under the supervision of J.F. and G.S.

## Competing interests

The authors declare no competing interests.
