## [Peer review file · Nature Communications]

REVIEWER COMMENTS

Reviewer #2 (Remarks to the Author):

The manuscript entitled “An ultrastrongly coupled single THz meta-atom” by S. Rajabali et al. investigates an individual metallic resonator coupled to inter-Landau level transitions via a novel setup. Overall, the manuscript is well written, properly detailed and technically sound. The three novelties of this work appear to be: i) technical: an asymmetric immersion lens setup (an extension of a symmetric version already utilized to investigate single quantum dots in the visible range) combined with a complementary metallic resonator design that allows performing THz spectroscopy of strongly subwavelength individual elements on a substrate; ii) the observation of quenching of the superradiance decay by decreasing the number of metallic resonators (something already observed in other studies, but not down to the single resonator limit has done here); iii) the characterization of a single meta-atom ultrastrongly-coupled to inter-Landau level transitions, leading to the observation of a coupling strength for the single resonator equivalent to the one measured for the array case, yet with a higher cooperativity (thanks to the superradiance quenching).

Even if the study does not lead to transformative/unexpected observations, is of a certain relevance for the scientific community, especially because it reports on a useful tool that can allow direct investigations on subwavelength systems (such as 2D-material flakes) via THz far-field spectroscopy, without requiring complex near-field implementations. For this reason, I recommend its publication in Nature Communications after a minor revision (see comments below).

- lines 166-167: “The measurement shows about 5 times signal enhancement in the time domain for the single resonator.” This signal enhancement is a bit obscure. Are the authors referring to the peak value of the THz waveform here? This should be clarified, adding perhaps the time-domain traces in the supplementary information file.

- lines 225-228: “The lower polariton (LP) mode of the coupled single resonator at its asymptotic limit, at $B = 4T$, has a Q-factor of 15.4 (using time trace decay method). The Q-factor of the LP at $B = 4T$ for the sample with the coupled 2D array of 3600 resonator without lenses and with lenses are 3.4 and 6.9, respectively”. It is really hard to see this directly from the results shown in Fig. 4 (indeed, the linewidth of the LP modes does not seem to be that different for the two cases in Fig. 4b). Since it is an important point, the authors should show more details regarding this Q-factor estimation in the supplementary information file.

- lines 238-242: “It is also evident that there is a renormalization of the loaded cavity frequency that blueshifts the resonance by 30 GHz ($\sim 0.1 \omega$) when we reduce the number of resonators from 3600 to 1 ... We do not observe such a shift when we investigate the cold cavity at 300 K (see Fig.3(a)).” My

impression is that a comparable slight blue-shift can also be seen in Fig. 3a, at least between the single resonator case and some of the other curves. Perhaps, this could be seen better if presented on a linear scale plot.

- Caption of Fig. 4b: “The distance between peaks at $B = 0.8T$ which is twice of the vacuum Rabi splitting is marked with purple” Shouldn’t this distance be the vacuum Rabi splitting directly? Ω should be defined in the text (is it the coupling strength, and if so, how is it different from the quantity g used in the cooperativity formula).

- lines 251-252: “The linewidth of the cyclotron resonance is smaller than the one measured without lenses due to the superradiance effect”. This is not very clear to me. How superradiance quenching can occur here just by varying the spot size of the illuminated area on the 2DEG? Is the density of emitters varying in this case?

- line 282: The cited Figures here are Fig 5c, 5d, not Fig. 4c, 4d.

Reviewer #3 (Remarks to the Author):

Rajabali et al. report on ultrastrong coupling of a single resonator and Landau level transitions in GaAs and InSb quantum wells using terahertz time-domain spectroscopy (THz TDS). The quenching of the Dicke superradiance has also been shown with respect to decreasing number of resonators. To achieve measurement sensitivity in THz TDS on a single resonator level, the authors utilized an asymmetric silicon immersion lens configuration. These results are novel and of interest to the broad community. Indeed, the combination of the asymmetric lens with THz TDS is even more general, as pointed out by the authors. The manuscript sounds technically correct and well written. Therefore, I recommend its publications in Nature Communications.

Few minor comments:

- Figure 1. “THz” should be added in front of TDS.
- The font size in Figure 1(e) is too small
- line 282: (Fig. 4c, 4d) should be replaced with (Fig. 5c, 5d)
- References style is inconsistent

Rebuttal Letter for Nature Communications manuscript NCOMMS-21-41287

Shima Rajabali^{1,*}, Sergej Markmann¹, Elsa Jöchl¹, Mattias Beck¹,
Christian A. Lehner², Werner Wegscheider², Jérôme Faist¹, Giacomo Scalari^{1,*}

¹ Institute of Quantum Electronics, ETH Zürich, 8093 Zürich, Switzerland

² Laboratory for solid state physics, ETH Zürich, 8093 Zürich, Switzerland

Please find here below our point-by-point reply to the Reviewers' questions. We would like to thank both Reviewers for their positive comments on our work, their in-depth reading of the manuscript and for their constructive criticisms which surely raised the readability and the overall scientific quality of our paper. We implemented several changes both in the main text and in the supplementary material. The point-to-point reply to the Reviewers' criticisms section is followed by another section that states the length of the text, methods, and legends and highlights the changes in the main text and in the supplementary material.

1 Point-by-point reply to the Reviewers' comments

Reviewer #1:

The manuscript entitled "An ultrastrongly coupled single THz meta-atom" by S. Rajabali et al. investigates an individual metallic resonator coupled to inter-Landau level transitions via a novel setup. Overall, the manuscript is well written, properly detailed and technically sound. The three novelties of this work appear to be: i) technical: an asymmetric immersion lens setup (an extension of a symmetric version already utilized to investigate single quantum dots in the visible range) combined with a complementary metallic resonator design that allows performing THz spectroscopy of strongly subwavelength individual elements on a substrate; ii) the observation of quenching of the superradiance decay by decreasing

the number of metallic resonators (something already observed in other studies, but not down to the single resonator limit has done here); iii) the characterization of a single meta-atom ultrastrongly-coupled to inter-Landau level transitions, leading to the observation of a coupling strength for the single resonator equivalent to the one measured for the array case, yet with a higher cooperativity (thanks to the superradiance quenching). Even if the study does not lead to transformative/unexpected observations, is of a certain relevance for the scientific community, especially because it reports on a useful tool that can allow direct investigations on subwavelength systems (such as 2D-material flakes) via THz far-field spectroscopy, without requiring complex near-field implementations. For this reason, I recommend its publication in Nature Communications after a minor revision (see comments below).

Question 1:

lines 166-167: “The measurement shows about 5 times signal enhancement in the time domain for the single resonator.” This signal enhancement is a bit obscure. Are the authors referring to the peak value of the THz waveform here? This should be clarified, adding perhaps the time-domain traces in the supplementary information file.

Figure R1: THz signal in time domain from a single resonator (cold cavity) without and with aSIL configuration. The delay between the two signals corresponds to the optical thickness of the aSIL assembly.

Answer: We thank the Reviewer for this question. We would like to clarify it in more detail to avoid confusion. The signal enhancement, as the Reviewer mentions, refers to the peak-to-peak value of the THz waveform for the single resonator. It is calculated by dividing the peak-to-peak value of the THz waveform with the aSIL configuration by the peak-to-peak value of the THz waveform without the aSIL assembly. For more clarity, Fig. R1 is added to the supplementary (Fig. S3).

Question 2:

lines 225-228: “The lower polariton (LP) mode of the coupled single resonator at its asymptotic limit, at $B = 4T$, has a Q-factor of 15.4 (using time trace decay method). The Q-factor of the LP at $B = 4T$ for the sample with the coupled 2D array of 3600 resonator without lenses and with lenses are 3.4 and 6.9, respectively”. It is really hard to see this directly from the results shown in Fig. 4 (indeed, the linewidth of the LP modes does not seem to be that different for the two cases in Fig. 4b). Since it is an important point, the authors should show more details regarding this Q-factor estimation in the supplementary information file.

Figure R2: (a) Normalized THz signal in time domain from a single cSRR and a 60×60 array of resonators with the aSIL configuration. The inset shows the magnified image of the tail of the decayed signal to highlight the oscillations. (b) The waveforms in panel (a) are digitally filtered with an ideal bandpass filter (centered at LP mode with a bandwidth ~ 400 GHz) and then fitted by sine functions with exponentially-decaying amplitudes to extract the oscillators’ decay times (τ_1, τ_2) and frequencies (f_1, f_2). The Q-factors of the LP mode at $B = 4T$ are then calculated as $Q_{single} = \frac{f_1}{\text{Linewidth}} = \frac{f_1}{1/\pi\tau_1} = \pi\tau_1 f_1 = \pi \times 16.3ps \times 300GHz = 15.4$ for the single resonator and $Q_{array} = \pi\tau_2 f_2 = \pi \times 7.87ps \times 280GHz = 6.9$ for the 60×60 array of resonators with aSIL configuration.

Answer: We agree with the Reviewer that the difference in Q-factors is not noticeable in Fig. 4. To acquire the colormaps in Fig. 4, we cut the time domain signal before the reflection signal from the cryostat windows and compute its Fourier transform. For a delay of 30 ps, the linewidth of the LP mode is then resolution limited. In order to calculate the

correct Q-factor for the LP at its asymptotic limit ($B = 4T$), we first cut the signal in the time domain before the echo. Then, the signal is digitally filtered by an ideal band-pass filter centered at the peak of the LP mode (bandwidth ~ 400 GHz) to remove the effect of the second mode. In the end, the filtered signal is fitted with a sine function with exponentially-decaying amplitude ($y = y_0 + A \exp(t/\tau) \sin(2\pi ft + \theta)$) to extract the oscillators' decay time (τ) and frequency (f) [1].

To clarify this point, Fig. R2 is added to the supplementary (Fig. S7).

Question 3:

lines 238-242: “It is also evident that there is a renormalization of the loaded cavity frequency that blueshifts the resonance by 30 GHz ($\sim 0.1 \omega$) when we reduce the number of resonators from 3600 to 1 . . . We do not observe such a shift when we investigate the cold cavity at 300 K (see Fig.3(a)).” My impression is that a comparable slight blue-shift can also be seen in Fig. 3a, at least between the single resonator case and some of the other curves. Perhaps, this could be seen better if presented on a linear scale plot.

Answer: We agree with the Reviewer that there is also a blue-shift in the cold cavity measurements (Fig. 3a) when we reduce the number of resonators. However, it is much smaller than the blue-shift in the loaded cavity (30 GHz). We attribute the larger shift in the case of the loaded cavity to the substantial dielectric contribution of the 2DEG that effectively produces a slow light effect. In the intermediate cases (the smaller array of resonators with more than one resonator), the resonant frequencies show larger shifts in Fig. 3a (as the Reviewer has mentioned). This larger shift can be because of the existence of other modes as a result of the interaction between the THz beam and the boundary of the resonator array : the description of these cases goes beyond the scope of this paper. Hence, we added Fig. R3 to the supplementary (Fig. S5) and modified the manuscript for more clarity:

“It is also evident that there is a renormalization of the loaded cavity frequency that blueshifts the resonance by 30 GHz ($\sim 0.1\omega$) when we reduce the number of resonators from 3600 to 1 . . . We observe a much smaller shift (7 GHz) when we investigate the cold cavity at 300 K (see Fig.3(a) and also Fig. S4 in the Supplementary info). We attribute the larger shift in the case of the loaded cavity to the substantial dielectric contribution of the 2DEG that effectively produces a slow light effect, enhancing the collective Lamb shift of the ensemble of meta atoms as observed in other systems.”

Question 4:

Caption of Fig. 4b: “The distance between peaks at $B = 0.8T$ which is twice of the vacuum Rabi splitting is marked with purple” Shouldn't this distance be the vacuum Rabi splitting directly? Ω should be defined in the text (is it the coupling strength, and if so, how is it different from the quantity g used in the cooperativity formula).

Answer: We agree with the Reviewer and apologize for the inconsistency in the text. The distance in Fig. 4b is vacuum Rabi splitting which is twice of the coupling strength. Ω is the coupling strength, noted also as g . We have modified the text accordingly and replaced

Figure R3: Cold cavity resonance measurement in Menlo (linear scale) for the single resonator and 60×60 array of resonators. Each curve is normalized to its peak value for a better comparison.

g with Ω .

Question 5:

lines 251-252: “The linewidth of the cyclotron resonance is smaller than the one measured without lenses due to the superradiance effect”. This is not very clear to me. How superradiance quenching can occur here just by varying the spot size of the illuminated area on the 2DEG? Is the density of emitters varying in this case?

Answer: In this case we are discussing the broadening of the cyclotron transition without any cavity. It has been proposed theoretically and proved experimentally that the cyclotron transition in 2D electron gases is superradiantly broadened: such research is reported in the paper by Zhang et al., [2].

Intersubband transitions have also been proven to undergo superradiant broadening [3]: an anomalous broadening is observed as a function of the angle of observation of the emission and attributed to an angular-dependent , superradiant dominated spontaneous emission of the electron plasma.

What we believe that is happening in the case of lens-coupled cyclotron measurement is that the focusing of the THz radiation and the successive out-coupling via immersion lenses effectively reduce the coupling of the electrons to the electromagnetic field. The distribution of the focused wave vectors implies an angle-dependent coupling of the THz electric field to the in-plane polarized cyclotron transition. In this respect we would expect a result similar to what obtained in Ref.[3] as a function of the extraction angle of the radiation due to the quantum well emission. We modified the manuscript to clarify this point and we plan to investigate further this aspect in the future.

Question 6:

line 282: The cited Figures here are Fig 5c, 5d, not Fig. 4c, 4d.

Answer: The Figure numbers are changed in the text.

Reviewer #2:

Rajabali et al. report on ultrastrong coupling of a single resonator and Landau level transitions in GaAs and InSb quantum wells using terahertz time-domain spectroscopy (THz TDS). The quenching of the Dicke superradiance has also been shown with respect to decreasing number of resonators. To achieve measurement sensitivity in THz TDS on a single resonator level, the authors utilized an asymmetric silicon immersion lens configuration. These results are novel and of interest to the broad community. Indeed, the combination of the asymmetric lens with THz TDS is even more general, as pointed out by the authors. The manuscript sounds technically correct and well written. Therefore, I recommend its publications in Nature Communications.

Few minor comments:

Comment 1:

Figure 1. “THz” should be added in front of TDS.

Answer: We thank the Reviewer. THz is added to the caption of Fig. 1.

Comment 2:

The font size in Figure 1(e) is too small

Answer: The font size is changed in Fig. 1e.

Comment 3:

line 282: (Fig. 4c, 4d) should be replaced with (Fig. 5c, 5d)

Answer: The Figure numbers are changed in the text.

Comment 4:

References style is inconsistent

Answer: The References style is modified and is consistent in the new version.

References

- [1] Wang, X. *et al.* Direct measurement of cyclotron coherence times of high-mobility two-dimensional electron gases. *Opt. Express* **18**, 12354–12361 (2010).
- [2] Zhang, Q. *et al.* Superradiant decay of cyclotron resonance of two-dimensional electron gases. *Phys. Rev. Lett.* **113**, 047601 (2014).

- [3] Laurent, T. *et al.* Superradiant emission from a collective excitation in a semiconductor. *Phys. Rev. Lett.* **115**, 187402 (2015).

2 The article information and list of changes

In this section, we provided information about the length of the main text and the supplementary document. We also added a new version of the article and supplementary where the changes are highlighted. The deleted parts are crossed out with red lines and the added parts are indicated with blue. The new references are also highlighted in yellow.

The length of the text, excluding “Methods”, “References”, and figure legends, is 4204 words. The “Abstract” section is 149 words. “Methods” section has two subsections and is 273 words. The article includes 5 figures. The length of their legends, excluding their standalone titles, are:

- Fig. 1 legend: 79 words
- Fig. 2 legend: 147 words
- Fig. 3 legend: 114 words
- Fig. 4 legend: 155 words
- Fig. 5 legend: 155 words

The article has 52 references. The supplementary document is 756 words, including legends and references (2 references) and has 8 figures.

A list of changes in the figures of the main text:

- Fig. 1: As Reviewer 2 asked, the font size in panel (e) is changed.
- Fig. 2: We lettered each panel and adjusted the caption. A magnified view is added to panel (c) for more clarity. The color bars in panel (g) and (h) are modified for clarity.
- Fig. 3: A frame is added around the plots for consistency.

An ultrastrongly coupled single THz meta-atom

Shima Rajabali^{1,*}, Sergej Markmann¹, Elsa Jöchl¹, Mattias Beck¹, Christian A. Lehner², Werner Wegscheider², Jérôme Faist¹, Giacomo Scalari^{1,*}

¹Institute of Quantum Electronics, ETH Zürich, 8093 Zürich, Switzerland

²Laboratory for solid state physics, ETH Zürich, 8093 Zürich, Switzerland

Free-space coupling to ~~strongly~~ subwavelength individual optical elements is a central theme in quantum optics, as it allows ~~to control and manipulate the properties of control over individual~~ quantum systems. ~~In this work, Here~~ we show that, by combining an asymmetric immersion lens setup and ~~a complementary design of metasurfaces resonating metasurface~~ we are able to perform THz ~~time domain time-domain~~ spectroscopy of an individual, strongly subwavelength ($\frac{d}{\lambda_0} = 1/20$) meta-atom. We unravel the linewidth dependence ~~of planar metamaterials~~ as a function of the meta-atom number indicating quenching of the ~~Dieke superradiancesuperradiant coupling~~. On these grounds, we investigate ultrastrongly coupled Landau polaritons at the single resonator level, measuring a normalized coupling ratio ~~of~~ $\frac{\Omega}{\omega} = 0.60$ ~~resulting from coupling of the fundamental mode to a few thousand electrons~~. Similar measurements on a ~~low loss, less doped lower density~~ two dimensional electron gas yield a coupling ratio $\frac{\Omega}{\omega} = 0.33$ with a cooperativity $C = \frac{4g^2}{\kappa\gamma} = 94$ ~~$C = \frac{4\Omega^2}{\kappa\gamma} = 94$~~ . ~~Interestingly, the coupling strength of a coupled single resonator is the same as of a coupled array~~. Our findings pave the way towards the control of ~~ultrastrong~~ light-matter interaction ~~in the ultrastrong~~

21 **coupling regime** at the single electron/ **single** resonator level. The proposed technique is way
**more general and can be useful to characterize the complex conductivity of micron-sized**
**samples in the THz ~~and sub-THz~~ domain.**
Extreme electronic and photonic confinement has recently allowed a number of groundbreak-
ing advances in several fields, from fundamental studies ¹ to applications ^{2,3}. Particularly interest-
ing is the possibility to manipulate *on-chip* the light-matter-coupling achieving new quasi-particles
called cavity polaritons ⁴ that profoundly modify the electro-optical properties of the constituent
elements.
Cavity polaritons rely on enhanced vacuum field fluctuations \mathcal{E}_0 to reach large values of the
vacuum Rabi frequency $\Omega_R \propto \mathcal{E}_0 \Omega \propto \mathcal{E}_0$ quantifying large values of coupling ⁵. The dependence
~~from~~ on the inverse of the cavity volume $\mathcal{E}_0 \propto 1/\sqrt{V}$ led to the development of strongly sub-
wavelength structures to reach the strongest couplings to-date ^{6,7}. Metallic-based cavities have
been used in the mid-infrared (Mid-IR)⁸ and terahertz (THz)⁹ regions of the electromagnetic spec-
trum for strong coupling experiments. Due to their subwavelength nature, the experiments have
been carried out on multiple cavities (100 – 1000 or more) in order to enhance the signal-to-
noise ratio (SNR). Since the frequency components excited for subwavelength features cannot
propagate in the far-field, only recently near-field techniques proved to be successful in probing
single resonator-based strongly coupled systems at Mid-IR frequencies at room temperature ^{10,11}.
In the field of ultrastrong coupling, the development of the Landau polariton platform¹² has led to
very ~~fascinating results~~ interesting developments, including studies of non-linear phenomena ¹³, the
reaching of high cooperativity ¹⁴ and the study of magnetotransport ¹⁵ in ~~ultrastrongly~~ ultrastrongly
coupled Hall systems ¹⁶. In this approach, extremely subwavelength planar split-ring resonators
have been used in combination with one-dimensionally confined semiconductor heterostructures
to achieve normalized coupling values $\frac{\Omega}{\omega} > 1.4$ (Ref ¹⁷).
The light-matter interaction strength in most of the solid-state-based cavity quantum electro-
dynamics (QED) systems scales with \sqrt{N} ~~with~~, N being the electron number due to the collective
(Dicke) enhancement of the interaction strength¹⁸. In Landau polaritons, when employing meta-
surface cavities, the number of coupled electrons is shared among several hundreds of identical
cavities. In the last few years, there has been a growing interest in studying ultrastrong light-matter
coupled systems towards the limit of few electrons coupled to a single cavity ¹⁹⁻²¹ to study the
fermionic Rabi model for the coupled systems rather than the bosonic Hopfield description where
both material excitation and the electromagnetic field are described as boson fields. Moreover,
such systems ~~can~~ could be building blocks for performing quantum information tasks. In this per-
spective a reduction in the number of coupled carriers towards the low limit of a few and ultimately
a single coupled element constitutes a great challenge both in terms of measurement sensitivity ^{1,22}
and optical cavity design.
The possibility and the limits on the interaction of free propagating far-field optical beams
from individual elements (molecules, atoms, quantum dots etc...) have been investigated both
theoretically ^{23,24} and experimentally ^{25,26}, mainly at visible or infrared frequencies. In our case
the individual element is not a quantum object but rather a single-subwavelength metallic planar
resonator, operating at millimeter wavelengths, lying in-between free space propagating beams and
guided microwaves. The single resonator spectroscopy at sub-THz/THz spectrum is performed by
implementing a simple and practical asymmetric Silicon immersion lens (aSIL) configuration. We
also demonstrate the linear dependence of the cold cavity linewidth of a 2D array of the metasurface
on the number of resonators in the array when the illuminated area with resonators becomes com-
parable or smaller than the wavelength. This observation ~~is an indication of quenching~~ indicates
the quenching of the superradiance decay. Moreover, the Landau polaritons in a coupled single-
subwavelength resonator are resolved on two different semiconductor heterostructures, and high
normalized coupling ~~strength~~ strengths of up to 60% and high cooperativities of up to $C = 94$ are
achieved.
**Measurements and results**
**Single resonator detection and ~~superradiance~~ linewidth** Free-space probing of a sample con-
taining a few/single subwavelength resonators is difficult as it is intrinsically inefficient in exciting
the right electromagnetic modes and generally features a very low ~~signal-to-noise ratio~~ SNR. We
thus developed a strategy based on the employment of ~~the complementary metamaterial approach~~
a complementary metamaterial combined with an aSIL configuration to optimize the matching of
a free space propagating beam to a single element of a complementary metasurface. The con-
ventional THz-time domain spectroscopy (TDS) setup we employ has a pair of off-axis parabolic
mirrors first to collimate and then focus the incident THz beam from the photo-conductive switch²⁷
on the sample. ~~The~~ Afterwards, the transmitted signal from the sample will be ~~then~~ collected, col-

[revised manuscript text omitted]

the situation is reversed as the transmitted signal is very low, but the only signal detected is the rel-
evant one. Fig. 2 visualizes the reasons that single resonator spectroscopy is possible using an aSIL
configuration. The front ~~and back hyperhemispherical Si lenses~~ hyperhemispherical Si lens with a
refractive index of $n_r = 3.42$ at $300GHz$ ³¹ ~~focuses the beam onto the resonator plane. The front~~
~~lens enhances the intensity of the beam arriving at the resonator plane (Fig. 2a) and both~~ permits to
optimally focus thanks to high numerical aperture and high refractive index. Both front and back
hyperhemispherical lenses expand the accessible wave vectors for far-field propagation ~~(Fig. 2e).~~
The back lens improves the collection of the signal and increases the frequency resolution due
to impedance matching and suppression of the echos refractive index matching with the sample
substrate and consequent suppression of Fabry-Pérot resonances (echoes in the time domain) from
the interface between the back of the sample and the back lens.
In Fig.2 we clarify, using a Fourier optics argument and some experimental measurements,
the mechanism at the basis of the different signals observed when probing a metasurface with a
large number of ~~resonator resonators~~ (Fig.2a) and a single meta-atom with a plane wave (Fig.2b)
compared to the case of a single meta-atom illuminated with an immersion lens ~~-(Fig.2c).~~
meta-atoms constituting the planar metamaterial are strongly subwavelength by design: they can
be usually probed by far-field optics because the near-field components diffracted by each element
are scattered in the far-field by the surrounding elements of the array. The unit cell dimension d_{unit}
is highly subwavelength as well (in our case $d_{unit} = 70 \mu m$ for a free space wavelength $\lambda_{fs} \simeq$
1 mm , with $\frac{d_{unit}}{\lambda_{fs}} = 7 \times 10^{-2}$) allowing the spatial frequencies to propagate (as represented in
Fig. 2a and 2b)its corresponding experimental far-field spectrum in Fig. 2d, as usually observed in
complementary metamaterials³². The presence of the resonator array is as well required for the
excitation of the resonant mode as the nearby elements diffract the incoming wave providing some
in-plane components of the wave vectors. In the case where the plane wave illuminates a single
subwavelength meta-atom (Fig. ~~2e~~2b), a large portion of spatial frequencies remain trapped in
the resonator's near field and do not reach the far-field detector. The corresponding experimental
spectrum does not show the resonant feature at 300 GHz ~~-(Fig.2e)~~. When the aSIL arrangement
is used (Fig. ~~2d~~2c), the spread in the wave vector excites the correct resonator mode that can be
then collected ~~from~~ by the back lens ~~and to~~ recover the searched transmission resonance ~~-(as~~
experimentally shown in (Fig.2f). In the lower row of ~~the figure Fig. 2,~~ we plot the near-field
~~electric field distribution~~ distribution of the electric field (Fig. 2g) and its spatial Fourier transform
~~-(The circles (Fig. 2h))~~. The circles in Fig. 2h, correspond to the maximum amplitude of the wave
vectors propagating in the ~~case of~~ cases without lenses (blue) and with lenses (red). It is evident that
~~the quasi-totality with the lenses~~ the quasi-totality of wave vectors can propagate in the far-field.
As a further confirmation of our model, we can cite the results obtained with a near-field probe
measuring a similar complementary single resonator²². In that experiment, the single resonator
was excited with a focusing lens, providing the necessary in-plane ~~k's~~ wave vectors. The resonant
signal was found to remain trapped in the near field since there was no collecting lens. Now that
we clarified the mechanism that underlies the observed transmission spectra, we can discuss the
experimental measurements in detail.
As a first step, we perform a systematic study of the cold cavity, illuminating a series of
samples deposited onto a semi-insulating GaAs substrate. These samples contain a decreasing
amount of resonators, from 3600 down to a single one. We study the linewidth and the transmission
amplitude of the cold cavity and their dependency on the number of resonators. The theory of the
superradiance ^{18,33–35} predicts a reduction in the collective radiative decay and consequently an
improvement in quality factor by reducing the number of two-level emitters confined in a spatial
region comparable or smaller than the wavelength. This has also been investigated experimentally
on metamaterials but never in the single resonator limit ³⁶. In order to study this effect, we designed
and fabricated 2D array of $n \times n$ complementary split-ring resonators (cSRR), including a very
large array of 60×60 resonators and small arrays with a varying n from 6 to 1, operating at a
resonant frequency of 350 GHz and a periodicity of $70 \mu m$ on a GaAs substrate (Fig. 1b).
We measured the resonators with a commercial THz spectrometer (details in Methods sec-
tion) at room temperature. We conducted the measurements with and without the back-to-back
aSIL system . As expected from previous research²², the resonant peak of the 2×2 cSRR array
and the single resonator could not be resolved without the use of our aSIL arrangement because
of the weak interaction of a few ~~resonator-resonators~~ with the THz beam (see the supplementary
infoSupplementary Material, Fig. S2). Even the measurement with 9 resonators shows a very weak
resonant peak. In Fig. 3, we report the result of the study, showing the collected spectra using the
aSIL assembly and the extracted quality factors (Q-factor) for ~~measurement~~ the measurements with
lenses. The measurement shows about 5 times signal enhancement in the peak-to-peak value of
the THz waveform in the time domain for the single resonator ~~-(see Supplementary Material, Fig.~~
S3). Moreover, the resonant peak for the arrays with less than 9 resonators were resolved with a
large dynamic range ($60dB$ for the single resonator).
Another key feature of this technique is the significant improvement in frequency resolution.
Since Si and GaAs have very close refractive ~~indexes~~ indices up to 1THz ³¹, the echoes of the
188 THz pulse, reflected at each interface, are greatly suppressed due to the impedance matching and,
as a consequence, the etalon effects are minimized. Particular care was taken to assemble the
stack lens-sample-lens without any air gap, pressing on the assembly with metallic positioners (~~see~~
~~Supplementary material~~). In the best conditions, the resolution was 18GHz (corresponding to a
time scan length of 55ps), limited by the reflection from the detector. If we compare the results of
the measurements on the large arrays with and without the lenses we observe a difference in the
resonant frequency due to the presence of the front Si lens. The presence of an high index material
in the near field of the complementary resonators red-shifts the resonance to values below 300GHz
which is also confirmed by our finite element simulations (more information in the ~~supplementary~~
~~document~~ Supplementary Material, Fig. S4).
To calculate the ~~Q-factor~~ Q-factor, we fitted the resonant peak with a Breit-Wigner-Fano
(BWF) function. Fig.3 shows an almost constant Q-factor for 3600 to 16 resonators measured with
lenses. For the samples with ~~a fewer number of~~ fewer resonators ($n \leq 3$), ~~as~~ the dimension of the
resonator field becomes comparable to the wavelength (at 300GHz , the wavelength propagating
in GaAs, $\lambda_n = \lambda/n_{\text{GaAs}} = 278\mu\text{m}$ and the one propagating in Si $\lambda_n = \lambda/n_{\text{Si}} = 292\mu\text{m}$), causing
the quenching of superradiance ~~emerges to~~ emerge. The Q-factor doubles from 9 resonators to a
single resonator ~~due~~. We attribute this Q-factor enhancement to the quenching of the superradiant
decay ~~and rather than to inhomogeneous broadening that is quite unlikely in this kind of THz~~
resonators. The Q-factor reaches ~ 11 for the single one. This can have important consequences
in the context of the strong coupling experiments that we will discuss in the next section. If we
compare our results with similar studies performed on arrays of metamaterials down to a few
elements³⁷, we observe a similar behaviour as in the case of "incoherent" resonators that are not
coupled by magnetic interaction, since the symmetric arrangement of the inductors in our case
cancels out the magnetic response. What ~~is clear in~~ emerges from our data is the enhancement of
the ~~Q-factor when the occupied area~~ Q-factor when the area occupied by the resonators is smaller
than λ^2 . It is worth noting that the number of illuminated resonators in case of the large array
(60×60 resonator) are 2628 for the measurement without lenses (a $4mm$ diameter illuminating
spot) and 44 for the measurement with aSIL configuration (a $500\mu m$ diameter illuminating spot).
**Ultrastrong coupling in a single complementary resonator** After having characterized the mea-
surement technique and studied the Q-factor dependence of the cSRR array on the number of res-
onators, we apply the developed method to the spectroscopy of a single resonator ultrastrongly
coupled to ~~an~~ inter-Landau level (LL) transitions in a single GaAs QW. With a similar fabrication
process, a large array and a single cSRR were deposited on top of a two dimensional electron
gas (2DEG) produced in a single GaAs square QW located $90nm$ below the surface. The sample
transmission is then measured in a THz-TDS setup at cryogenic temperature ($T = 2.7K$) as a
function of magnetic field swept between 0 and $4T$. The best resolution for this setup is $33GHz$
(corresponding to a time scan length of $30ps$), limited by the thickness of the cryostat windows.
Further details are to be found in the Methods section. In order to gain insight into the coupling of
the complete metasurface and the single resonator to the THz beam, we measured the transmission
spectra for both ~~kind~~ kinds of samples with and without the aSIL assembly. The results, displayed

Figure 3: **Quenching of the superradiance decay** (a) Transmittance of $n \times n$ array of cSRRs ($n = 1$ to 6) and a large array with $n = 60$. These measurements are done using the aSIL assembly and the resonant peak of the single resonator is resolved. (b) is **Q-factor** vs. the number of resonators extracted from the measurements with lenses. The color of the data points are the same as the color of the transmittance curves in panel (a). The blue dashed line shows the linear dependence of the quality factor on the density of the resonators ($Q = \frac{6.04}{N} + 4.6$). The Q-factors are calculated **using-by** fitting the resonant peak with **BWF-Breit-Wigner-Fano** function. All error bars represent one σ confidence interval.

as transmittance color maps as a function of the magnetic field, are reported in Fig. 4. The figure
 shows a comparison between the transmission measurements without (top row) and with aSIL con-
 figuration (bottom row) for a 2D array of 60×60 resonators (left column) and a single resonator
 (right column). The colormaps relative to the plane wave (without lenses) case clearly show very
 well resolved polaritonic branches for the 3600 resonator sample and a very different spectrum in
the case of a single resonator. In the single resonator measurement, we observe a broad spectral
feature corresponding to high transmission that extends from $180GHz$ to $300GHz$ and is inde-
pendent from the applied magnetic field. A narrow absorption feature linearly changing with the
magnetic field and corresponding to the cyclotron resonance ~~is crossing~~ crosses the broad trans-
mission peak. The single resonator excited by a plane wave, whose radiation is collected without
an immersion lens, ~~is not showing~~ does not show ultrastrong coupling since the correct resonant
LC mode cannot be excited and detected. The measurement in the case of the single resonator
measured with the aSIL assembly is significantly different: we observe extremely well resolved
polaritonic branches that compare well with ~~the one~~ those observed in the 3600 resonator sample
in both cases with and without the lenses. We can now analyze in detail and compare the polaritons
measured in the case of employing the aSIL assembly for the single resonator and the large array.
The lower polariton (LP) mode of the coupled single resonator at its asymptotic limit, at $B =$
$4T$, has a Q-factor of 15.4 (using time trace decay method, more information in the Supplementary
Material, Fig. S7). The Q-factor of the LP at $B = 4T$ for the sample with the coupled 2D array
of 3600 resonator without lenses and with lenses are 3.4 and 6.9, respectively. Similar to the cold
cavity in the previous section, the Q-factor of the LP at its asymptotic limit is higher for the single
resonator compared to the one for the array due to the quenching of the ~~superradiance~~ superradiant
decay. By extracting the maximum of the spectrum at each magnetic field and fitting them with
the Hopfield model⁵, a normalized coupling of $\frac{\Omega}{\omega} = 33\%$ is achieved for the single resonator
measured in a confocal configuration. For the array of 3600 resonators the normalized coupling
for the measurement with and without lenses are $\frac{\Omega}{\omega} = 32\%$ and 36% , respectively. The cross
sections of the colormap of the 60×60 array and the single resonator measurement with lenses at
 three different magnetic field values of $0T$, $800mT$, and $4T$ show the well resolved upper polariton
 (UP) and LP peaks in the single resonator measurement (Fig. 4b). ~~We notice that, as expected, the~~
 ~~polaritonic branches in the single resonator case display a higher quality factor with respect to the~~

[revised manuscript text omitted]
. (c) and (d): Sections of the colormaps in panel (a) and (b) are represented in panel (c) and (d), respectively, at different magnetic field values ($B = 0, 0.425T$ and $1.5T$). To have a more visible LP and UP, a section at $B = 0.425T$ which is at a marginally higher value than the anti-crossing is chosen.

To conclude, we presented a back-to-back Si immersion lens setup with an asymmetric con-
figuration allowing the spectroscopy of highly subwavelength individual THz cavities. Using this
platform, we resolved the far-field transmission measurements of an ultrastrongly coupled, sub-
wavelength split-ring single resonator to a LL transition in a single GaAs QW and a single InSb
QW. The highest coupling of 60% for only about 2000 coupled electrons is reported for a single
cSRR on a single InSb QW. As our results demonstrate, the combination of the aSIL configura-
tion with a complementary-based resonant metallic structure paves the way to single-object, highly
subwavelength spectroscopy of quantum electrodynamics systems operating in the mm-wave and
322 THz range. The proposed experimental scheme can be extended to the study of dynamical optical
conductivity of high-quality 2D structures (graphene, TMDC's, Van der Waals heterostructures)⁴⁷
with very small effective areas (i.e. $10 \times 10 \mu\text{m}^2$) resulting from exfoliation procedures.
The data that support the findings of this Article are available in the ETH Research Collection ⁴⁸.
**Methods**
**Asymmetric lens setup and sample fabrication** The lenses are hyperhemispherical and hemi-
spherical ones fabricated with high resistivity Silicon (Tydex⁴⁹) of diameter $2r_1 = 4\text{mm}$ and
$2r_2 = 8\text{mm}$, respectively. The ~~complementary sample layout is conceived in order to ease the~~
~~alignment of the front lens. The sample is~~ sample is metallized with a circular boundary with a
diameter of 4mm , matching the edge of the top lens (see Fig. 1c). The lens and the sample can be
then accurately aligned under the optical microscope. Mechanical clamps ensure a close contact of
the whole assembly front lens-sample-back lens, forming a quasi-index matched ($n_{Si}^{350\text{GHz}} = 3.42$,
$n_{GaAs}^{350GHz} = 3.52$) stack of total length $L_s \simeq 6.5mm$. The resonators were simulated using a com-

[revised manuscript text omitted]

467 **Correspondence** *Correspondence should be addressed to S. Rajabali (email: shimar@phys.ethz.ch) and
468 G. Scalari (email: scalari@phys.ethz.ch).

REVIEWERS' COMMENTS

Reviewer #2 (Remarks to the Author):

The authors have provided the requested information and have modified the manuscript accordingly. I recommend publication.

Luca Razzari

Reviewer #3 (Remarks to the Author):

The authors have addressed my comments. The manuscript can be published.